# Addressing "Documentation Debt" in Machine Learning: A Retrospective Datasheet for BookCorpus

**Jack Bandy**
Northwestern University
jackbandy@u.northwestern.edu

**Nicholas Vincent**
Northwestern University
nickvincent@u.northwestern.edu

## Abstract

This paper contributes a formal case study in retrospective dataset documentation and pinpoints several problems with the influential BookCorpus dataset. Recent work has underscored the importance of dataset documentation in machine learning research, including by addressing "documentation debt" for datasets that have been used widely but documented sparsely. BookCorpus is one such dataset. Researchers have used BookCorpus to train OpenAI's GPT-N models and Google's BERT models, but little to no documentation exists about the dataset's motivation, composition, collection process, etc. We offer a retrospective datasheet with key context and information about BookCorpus, including several notable deficiencies. In particular, we find evidence that (1) BookCorpus violates copyright restrictions for many books, (2) BookCorpus contains thousands of duplicated books, and (3) BookCorpus exhibits significant skews in genre representation. We also find hints of other potential deficiencies that call for future research, such as lopsided author contributions. While more work remains, this initial effort to provide a datasheet for BookCorpus offers a cautionary case study and adds to growing literature that urges more careful, systematic documentation of machine learning datasets.

## 1 Introduction

Large language models are "growing" in a number of ways: the volume of parameters in the models (e.g. 175 Billion in OpenAI's full GPT-3 [5]), the range of use cases (e.g. to help train volunteer counselors for the Trevor Project [33]), the degree to which these models affect the public (e.g. powering almost every English query on Google [41]), and crucially, the size and complexity of the text data used for training. Bender and Gebru et al. [2] suggest that training data currently faces "documentation debt," in that popular language models are trained on sparsely-documented datasets which are often difficult to replicate and comprehend.

One such sparsely-documented dataset is BookCorpus. Originally introduced by Zhu and Kiros et al. [49] in 2014, BookCorpus and derivative datasets have been used to train Google's influential "BERT" model [11] (amassing over 20,000 Google Scholar citations as of June 2021), BERT's variants such as RoBERTa [30] and ALBERT [26], OpenAI's GPT-N models [38], XLNet [48], and more. Yet researchers provide scant details about BookCorpus, often merely noting the number of books and tokens in the dataset, or the total disk space it occupies. When introducing the dataset in 2014, [49] provided six summary statistics (shown in Table 1) along with the following description:

> In order to train our sentence similarity model we collected a corpus of 11,038 books from the web. These are free books written by yet unpublished authors. We only included books that had more than 20K words in order to filter out perhaps noisier shorter stories. The dataset has books in 16 different genres, e.g., Romance (2,865 books), Fantasy (1,479), Science fiction (786), etc. Table [1] highlights the summary statistics of our corpus.

35th Conference on Neural Information Processing Systems (NeurIPS 2021), Sydney, Australia.

| # of books | # of sentences | # of words | # of unique words | mean # of words per sentence | median # of words per sentences |
|---|---|---|---|---|---|
| 11,038 | 74,004,228 | 984,846,357 | 1,316,420 | 13 | 11 |

Table 1: The six summary statistics of BookCorpus originally provided in Table 2 from [49]

This paper attempts to help address "documentation debt" for the widely-used BookCorpus dataset, answering a growing body of work that calls more careful dataset documentation in machine learning research (e.g. "dataset nutrition labels" [21, 22, 44], "data statements" [1], "dataset disclosure forms" [43], "datasheets" [20, 2], and other types of "accountability frameworks" [23]). Ongoing findings further underscore the importance of dataset documentation. For example, Northcutt et al. [32] found pervasive label errors in test sets from popular benchmarks used in computer vision (e.g. MNIST, ImageNet) and natural language processing (e.g. IMDB movie reviews, 20 Newsgroups, Amazon Reviews). In documenting the Colossal Clean Crawled Corpus (C4), Dodge et al.[13] found that the corpus contained more tokens from `patents.google.com` than from English Wikipedia.

Building on related work, this paper provides documentation for an unlabeled text dataset that has mainly been used in unsupervised settings (such as training large language models). While documentation efforts exist for datasets with similar influence (e.g. [13, 32]), to our knowledge, this paper contributes the first formal documentation effort for BookCorpus. To do so, we apply the datasheet framework described by [20] to retrospectively document the motivation, composition, and collection process for BookCorpus, as well as other aspects for researchers to consider when deciding if and how to use the dataset.

In addition to documenting important general information about BookCorpus such as its original use cases, overall composition, and sources of funding, we find several notable deficiencies in the dataset. For one, many books contain copyright restrictions that, through a plain language reading, would have prevented them from being distributed in BookCorpus and similar datasets. We also find that thousands of books in BookCorpus are duplicated, with only 7,185 unique books out of 11,038 total. Third, we find notable genre skews in the dataset, for example, romance novels are significantly over-represented compared to the newer BookCorpusOpen. In addition to these three deficiencies, we also find a number of *potential* deficiencies that motivate future work, such as lopsided author contributions, potentially skewed religious representation, and adult content that may be problematic in some use cases. We conclude with a discussion of future work, implications, and the value of systematic dataset documentation for machine learning research.

## 2   Background

A growing body of machine learning research recognizes the unique challenges and importance of datasets. In the words of Paullada et al. [35], "the importance of datasets for machine learning research cannot be overstated." Although prominent research tends to be model-focused, multiple reviews [34, 37] suggest that data management creates many challenges and takes up a substantial portion of many ML workflows, including dataset generation, preprocessing, and augmentation. Given the sizable role and impact of datasets, researchers have started advocating for more data-focused ML work, such as budgeting project time specifically for dataset management [2], and/or leveraging "MLOps" to collaborate on shared, high-quality datasets [40].

In some cases, data-focused ML research is proactive. For example, [36] introduced the AViD dataset for action recognition from videos. The authors show how AViD addresses shortcomings of YouTube-based datasets, illustrate its usefulness for pretraining models, and position it as a valuable dataset for a range of computer vision tasks in the future.

Data-focused ML research can also be retroactive, revisiting influential datasets that may be problematic or under-explored. The ImageNet dataset [10], for instance, has served as a key benchmark for ML researchers since 2009. However, [3] suggests the focus on ImageNet may have caused some overfitting to its idiosyncrasies: only one label per image, overly restrictive labels, and arbitrary distinctions between labels. ImageNet also contains pervasive label errors [32], including in the validation set (2,196 errors, or 6% of the set). [32] also found errors in other image datasets like MNIST and CIFAR, as well as text datasets such as IMDB movie reviews and 20 Newsgroups.

As language models like GPT and BERT receive more research attention, data-focused ML research has started exploring the text datasets used to to train them. For instance, [13] provides retrospective documentation for the Colossal Clean Crawled Corpus (C4, introduced by [39]), and [7] analyzes

several large-scale web-crawled text datasets. These retrospective analyses have provided fruitful insights for influential web-crawled text datasets, however, we are aware of no similar work documenting the influential BookCorpus dataset. Some informal blog posts [45] and journalistic analyses [27] have provided initial insights, but much remains unknown. We therefore aim to provide retrospective documentation for the BookCorpus dataset, equipping the ML community to make informed decisions as to if and how they should use BookCorpus in the future.

## 3 Methods

### 3.1 Documentation and Analysis

The authors systematically addressed all questions in the datasheet template by [20].[1] While we did not deem all questions relevant to BookCorpus, we still include these questions and note our reasoning. Furthermore, as encouraged by [20], we include some additional questions that are important for understanding and using BookCorpus. To distinguish between "official" datasheet questions from [20] and additional questions, we denote our additions with a *[+]* preceding the question.

### 3.2 Data Collection

To create this datasheet, we collected and analyzed three different versions of BookCorpus: (1) the original 2014 BookCorpus (collected from the authors' website [49]), (2) BookCorpusOpen [16] (a 2020 version included in "The Pile" [19]), and (3) Smashwords21 (a "superset" of all books listed on Smashwords, which we collected ourselves). Section A.1 contains more details about each dataset.

## 4 Datasheet Questions and Answers for BookCorpus

### 4.1 Motivation

*For what purpose was BookCorpus created?* BookCorpus was originally created to help train a neural network that could provide "descriptive explanations for visual content" [49]. Specifically, BookCorpus was used to train a sentence embedding model for aligning dialogue sentences from movie subtitles with written sentences from a corresponding book. After unsupervised training on BookCorpus, the authors' encoder model could "map any sentence through the encoder to obtain vector representations, then score their similarity through an inner product" [49].

*[+] For what purpose were the books in BookCorpus created?* The books in BookCorpus were self-published by authors on Smashwords, likely with a range of motivations. While it is safe to assume that authors publishing free books via Smashwords had some motivation to share creative works, there is no way to verify they were interested in training AI systems. For example, many authors in BookCorpus explicitly license their books "for [the reader's] personal enjoyment only," limiting reproduction and redistribution. When notified about BookCorpus and its uses, one author from Smashwords said "it didn't even occur to me that a machine could read my book" [27].

*Who collected BookCorpus?* BookCorpus was collected by Zhu and Kiros et al. [49] from the University of Toronto and the Massachusetts Institute of Technology. Their original paper includes seven authors, but does not specify who was involved in collecting BookCorpus.

*[+] Who created the books in BookCorpus?* BookCorpus' constituent data was created by a large number of self-published authors on Smashwords. These authors wrote the books and sentences that make up BookCorpus, and now support a wide range of machine learning systems.

*[+] How many people were involved in creating BookCorpus?* The original BookCorpus dataset does not include the structured metadata required to answer this question. However, here we provide an estimate based on the number of unique authors who contributed free books to Smashwords21. In Smashwords21, 29,272 unique authors contributed 65,556 free books, which included 1.77 billion total words. Assuming a similar ratio of authors to books holds for the subset of books that were free, and thus used in the original BooksCorpus, we estimate that about 3,490 authors were involved in creating the original dataset of 7,185 books (i.e. $(29,272/65,556)*7,815 = 3,489.5$).

---

[1]Replication materials are available at https://github.com/jackbandy/bookcorpus-datasheet

Notably, author contributions appear to be highly concentrated: among free books in Smashwords21, the top 10% of authors by word count were responsible for 59% of all words in the dataset, and the top 10% by book count were responsible for 43% of all books.

*Who funded the creation of BookCorpus?* The original paper by Zhu and Kiros et al. [49] acknowledges support from the Natural Sciences and Engineering Research Council (NSERC), the Canadian Institute for Advanced Research (CIFAR), Samsung, Google, and a grant from the Office of Naval Research (ONR). They do not specify how funding was distributed across these sources.

*[+] Who funded the books within BookCorpus?* Broadly, many authors on Smashwords do make money by selling ebooks to readers (including on other platforms like Kindle, Audible, Barnes and Noble, and Kobo), although many also write books as a hobby alongside other occupations. Some books in BookCorpus may have been commissioned in some way.

## 4.2 Composition

*What do the instances in BookCorpus represent?* BookCorpus consists of text files, each of which corresponds to a single book from Smashwords. Zhu and Kiros et al. [49] also provide two large files in which each row represents a sentence.

*How many instances (books) are there in total?* In the original dataset described by Zhu and Kiros et al. [49], BookCorpus contained **11,038** books. However, in the files we obtained, there are only **7,185 unique** books (excluding `romance-all.txt` and `adventure-all.txt` as explained in A.1.1). Section A.2 contains further details as to how we confirmed duplicate books.

*Does BookCorpus contain all possible instances (books) or is it a sample?* **Sample**. BookCorpus contains free books from Smashwords which are at least 20,000 words long. Based on metrics from Smashwords [9], 11,038 books (as reported in the original BookCorpus dataset) would have represented approximately 3% of the 336,400 books published on Smashwords as of 2014, while the 7,185 unique books we report would have represented **2%**. For reference, as of 2013, the Library of Congress contained 23,592,066 cataloged books [17].

*What data does each instance (book) consist of?* Each book in BookCorpus includes the full text from the ebook (often including preamble, copyright text, etc.). However, in research that uses BookCorpus, authors have applied a range of different encoding schemes that change the definition of an "instance" (e.g. in GPT-N training, text is encoded using byte-pair encoding).

*Is there a label or target associated with each instance (book)?* **No**. The text from each book was originally used for unsupervised training by Zhu and Kiros et al. [49], and the only label-like attribute is the genre associated with each book, which is provided by Smashwords.

*Is any information missing from individual instances (books)?* **Yes**. We found 98 empty book files in the `books_txt_full` directory from the original BookCorpus [49]. Also, while the authors collected books longer than 20,000 words, we found that 655 files were shorter than 20,000 words, and 291 were shorter than 10,000 words, suggesting that many book files were significantly truncated from their original text.

*Are relationships between individual instances (books) made explicit?* **No**. The data implicitly links books in the same genre by placing them in the same directory. We also found that duplicate books are implicitly linked through identical filenames. However, no other relationships are made explicit, such as books by the same author, books in the same series, books set in the same context, books addressing the same event, and/or books using the same characters.

*Are there recommended data splits?* **No**. The authors use all books in the dataset for unsupervised training, with no splits or subsamples.

*Are there any errors, sources of noise, or redundancies in BookCorpus?* **Yes**. While some book files appear to be cleaned of preamble and postscript text, many files still contain such text as well as various other sources of noise. Of particular concern is that we found many copyright-related sentences. For example, the sentence "this book remains the copyrighted property of the author, and may not be redistributed to others for commercial or non-commercial purposes..." occurred 111 times in the `books_in_sentences` files. Here, we note that this and similar sentences represent noise and redundancy, though we return to the issue of copyrights in Section 4.6.

Another source of noise is that, as previously noted, BookCorpus contains many duplicate books: of the 7,185 unique books in the dataset, 2,930 occurred more than once. Most of these books (N=2,101) appeared twice, though many were duplicated multiple times, including some books (N=6) with five copies in BookCorpus. See Table 2.

*Is BookCorpus self-contained?* **No**. Although Zhu and Kiros et al. [49] maintained a self-contained version of BookCorpus on their website for some time, there is no longer an official publicly-available version of the original dataset. We obtained the dataset from their website through a security vulnerability,[2] but the public web page for the project now states: "Please visit smashwords.com to collect your own version of BookCorpus" [49]. Thus, researchers who wish to use BookCorpus or a similar dataset must either use a new public version such as BookCorpusOpen [16], or generate a new dataset from Smashwords via "Homemade BookCorpus" [25].

Smashwords is an ebook website that describes itself as "the world's largest distributor of indie ebooks."[3] Launched in 2008 with 140 books and 90 authors, by 2014 (the year before BookCorpus was published) the site hosted 336,400 books from 101,300 authors [9]. As of 2020, it hosted 556,800 books from 154,100 authors [8].

*Does BookCorpus contain data that might be considered confidential?* **Likely no.** While we did find personal contact information in the data (discussed further below), the books do not appear to contain any other restricted information, especially since authors opt-in to publishing their books.

*Does BookCorpus contain data that, if viewed directly, might be offensive, insulting, threatening, or might otherwise cause anxiety?* **Yes**. While this topic may warrant further research, as supporting evidence, we found that 537,878 unique sentences (representing 984,028 total occurrences) in the `books_in_sentences` files contained one or more words in a commonly-used list of "Dirty, Naughty, Obscene, and Otherwise Bad Words" [14]. However, merely using one of these words does not constitute an offensive or insulting sentence. We inspected a random sample of these sentences, finding some fairly innocuous profanities (e.g. the sentence "oh, shit." occurred 250 times), some pornographic dialogue, some hateful slurs, and a range of other potentially problematic content. Section 5 further discusses how some sentences and books may be problematic for various use cases.

*Does BookCorpus relate to people?* **Yes**, each book is associated with an author.

*Does BookCorpus identify any subpopulations?* **No**. BookCorpus does not identify books by author or any author demographics, and the `books_in_sentences` directory even aggregates all books into just two files. The `books_txt_full` directory identifies 16 genres, though we do not consider genres to be subpopulations since they correspond to books rather than authors.

*Is it possible to identify individuals (i.e., one or more natural persons), either directly or indirectly (i.e., in combination with other data) from BookCorpus?* **Likely yes**. In reviewing a sample of books, we found that many authors provide personally-identifiable information, often in the form of a personal email address for readers interested in contacting them.

*Does the dataset contain data that might be considered sensitive in any way?* **Yes**. The aforementioned contact information (email addresses) is sensitive personal information.

*[+] How does the sample compare to the population in terms of genres?* Compared to BookCorpusOpen and all books on Smashwords, the original BookCorpus appears to have several dominant genres. This is to be expected given the filtering applied (only free books, longer than 20,000 words), although some aspects of the skew suggest further research may be helpful. See Table 3. We note that, of course, there is no "natural distribution" of book genres or gold standard to which the observed frequencies should be compared. Rather, we argue that the distribution of genre, religious content, etc. must be considered in the context of how a dataset will be used, and that documentation work will make doing so much easier.

*[+] How does the sample compare to the population in terms of religious viewpoint?* This question is currently impossible to answer using data from the original BookCorpus, however it is possible for BookCorpusOpen and Smashwords21. The metadata for these datasets only includes religions as subjects, not necessarily as viewpoints. For example, metadata might indicate a book is about Islam, though its author writes from an Atheist viewpoint.

---

[2]We have notified the authors of the security vulnerability that allowed us to download the dataset.
[3]https://www.smashwords.com/

Still, we did find notable skews in religious representation in Smashwords21 and BookCorpusOpen, hinting that BookCorpus may exhibit similar skews. Following the recently-introduced BOLD framework [12], we tabulated based on seven of the most common religions in the world: Sikhism, Judaism, Islam, Hinduism, Christianity, Buddhism, and Atheism. Overall, Smashwords appears to over-represent books about Christianity, and BookCorpusOpen over-represents books about Islam. See Table 4 in Section A.3

## 4.3 Collection Process

*How was the data associated with each instance (book) acquired?* The text for each book was downloaded from Smashwords.

*What mechanisms or procedures were used to collect BookCorpus?* The data was collected via scraping software. While the original scraping program is not available, replicas (e.g. [25]) operate by first scraping Smashwords to generate a list of links to free ebooks, downloading each ebook as an epub file, then converting each epub file into a plain text file.

*What was the sampling strategy for BookCorpus?* Books were included in the original BookCorpus if they were available for free on Smashwords and longer than 20,000 words, thus representing a non-probabilistic **convenience sample**. The 20,000 word cutoff likely comes from the Smashwords interface, which provides a filtering tool to only display books "Over 20K words."

*Who was involved in collecting BookCorpus and how were they compensated?* **Unknown**. The original paper by Zhu and Kiros et al. [49] does not specify which authors collected and processed the data, nor how they were compensated.

*Over what timeframe was BookCorpus collected?* **Unknown**. BookCorpus was originally collected some time well before the original paper [49] was presented at the International Conference on Computer Vision (ICCV) in December 2015.[4]

*Were any ethical review processes conducted?* **Likely no**. Zhu and Kiros et al. [49] do not mention an Institutional Review Board (IRB) or other ethical review process involved in their original paper.

*Does the dataset relate to people?* **Yes**, each book is associated with an author (thus determining that the following three questions should be addressed).

*Was BookCorpus collected from individuals, or obtained via a third party?* **Third party**. BookCorpus was collected from Smashwords, not directly from the authors.

*Were the authors notified about the data collection?* **No**. Discussing BookCorpus in 2016, Richard Lea wrote in *The Guardian* that "The only problem is that [researchers] didn't ask" [27]. When notified about BookCorpus and its uses, one author from Smashwords said "it didn't even occur to me that a machine could read my book" [27].

*Did the authors consent to the collection and use of their books?* **No**. While authors on Smashwords published their books for free, they did not consent to including their work in BookCorpus, and many books contain copyright restrictions intended to prevent redistribution (based on a plain language interpretation). As described by Richard Lea in *The Guardian* [27], many books in BookCorpus include:

> a copyright declaration that reserves "all rights", specifies that the ebook is "licensed for your personal enjoyment only", and offers the reader thanks for "respecting the hard work of this author"

Considering these copyright declarations, authors did not explicitly consent to include their work in BookCorpus or related datasets. Using the framework of consentful tech [28], a consentful version of BookCorpus would ideally involve author consent that is **F**reely given, **R**eversible, **I**nformed, **E**nthusiastic, and **S**pecific (FRIES).

*Were the authors provided with a mechanism to revoke their consent in the future or for certain uses?* **Likely no**. For example, if an author released a book for free before BookCorpus was collected, then changed the price and/or copyright after BookCorpus was collected, the book likely remained in BookCorpus. In fact, preliminary analysis suggests that this is the case for at least 323 unique books

---

[4]http://pamitc.org/iccv15/

in BookCorpus which are no longer free to download from Smashwords, and would cost $930.18 to purchase as of April 2021.

*Has an analysis of the potential impact of BookCorpus and its use on data subjects been conducted?*
**Likely no**. Richard Lea interviewed some authors represented in BookCorpus in 2016 [27], but we are not aware of any holistic impact analysis.

### 4.4 Cleaning and Labeling

*Was any labeling done for BookCorpus?* While the original paper by Zhu and Kiros et al. [49] did not use labels for supervised learning, each book is labeled with genres, which are supplied by Smashwords authors themselves.

*Was any cleaning done for BookCorpus?* **Likely yes**. The `.txt` files in BookCorpus seem to have been partially cleaned of some preamble text and postscript text, however, Zhu and Kiros et al. [49] do not mention the specific cleaning steps. Also, many files still contain some preamble and postscript text, including many sentences about licensing and copyrights. As another example, the sentence "please do not participate in or encourage piracy of copyrighted materials in violation of the author's rights" occurs 40 times in the BookCorpus `books_in_sentences` files.

Additionally, based on samples we reviewed from the original BookCorpus, the text was tokenized to some extent (e.g. contractions are split into two words).

*Was the "raw" data saved in addition to the cleaned data?* **Unknown**.

*Is the software used to clean BookCorpus available?* While the original software is not available, replication attempts (e.g. [25]) provide some software for turning `.epub` files into `.txt` files and subsequently cleaning them.

### 4.5 Uses

*For what tasks has BookCorpus been used?* BookCorpus was originally used to train a sentence embedding model for a system meant to provide descriptions of visual content (i.e. to "align" books and movies), but the dataset has since been applied in many different use cases. Namely, BookCorpus has been used to help train more than thirty influential language models as of April 2021 [15], including Google's enormously influential BERT model which was shown to be applicable to a wide range of language tasks (e.g. answering questions, language inference, translation, and more).

*Is there a repository that links to any or all papers or systems that use BookCorpus?* On the dataset card for BookCorpus [15], Hugging Face provides a list of more than 30 popular language models that were trained or fine-tuned on the dataset.

*What (other) tasks could the dataset be used for?* Given that embedding text and training language models are useful prerequisites for a huge number of language related tasks, BookCorpus could prove useful in a wide range of pipelines and English language tasks. However, as discussed below, this work highlights the need for caution when applying this dataset.

*Is there anything about the composition of BookCorpus or the way it was collected and preprocessed/cleaned/labeled that might impact future uses?* **Yes**. At minimum, future uses should curate a subsample of BookCorpus, being mindful of copyright restrictions, duplicate books, sampling skews, and other potential deficiencies noted in this datasheet.

*Are there tasks for which BookCorpus should not be used?* Our work strongly suggests that researchers should use BookCorpus with caution for any task, namely due to potential copyright violations, duplicate books, and sampling skews. These concerns also apply to derivative datasets, such as BookCorpusOpen, as they are prone to similar deficiencies given their reliance on Smashwords.

### 4.6 Distribution

*How was BookCorpus originally distributed?* For some time, Zhu and Kiros et al. [49] distributed BookCorpus from a web page. The page now states "Please visit smashwords.com to collect your own version of BookCorpus" [49].

*How is BookCorpus currently distributed?* While there have been various efforts to replicate Book-Corpus, one of the more formal efforts is BookCorpusOpen [16], included in the Pile [19] as "BookCorpus2." Furthermore, GitHub users maintain a "Homemade BookCorpus" repository [25] with various pre-compiled tarballs that contain thousands of books.

*Is BookCorpus distributed under a copyright or other intellectual property (IP) license, and/or under applicable terms of use (ToU)?* To our knowledge, the BookCorpus dataset has never stated any copyright restrictions, but the same is not true of books within BookCorpus.

In reviewing sources of noise in BookCorpus, we found 111 instances of the sentence, "this book remains the copyrighted property of the author, and may not be redistributed to others for commercial or non-commercial purposes." We also found 109 instances of the sentence "although this is a free book, it remains the copyrighted property of the author, and may not be reproduced, copied and distributed for commercial or non-commercial purposes," and hundreds of instances similar sentences. This makes clear that the the direct distribution of BookCorpus violated copyright restrictions for many books, at least from a plain language interpretation. In the past, some applications of fair use doctrine have provided legal protection for related cases in machine learning [29]. Still, it seems likely that the distribution and reproduction of BookCorpus may be legally fraught. Further work from copyright experts would be needed to clarify the nature of these violations and potential reconciliatory measures to compensate authors.

Relatedly, some books in BookCorpus now cost money even though they were free when the original dataset was collected. By matching metadata from Smashwords for 2,680 of the 7,185 unique books in BookCorpus, we found that 323 of these 2,680 books now cost money to download. The total cost to purchase these books as of April 2021 would be $930.18, and this represents a lower bound since we only matched metadata for 2,680 of the 7,185 books in BookCorpus.

*Have any third parties imposed restrictions on BookCorpus?* **No**, at least not to our knowledge.

*Do any export controls or other regulatory restrictions apply to the dataset or to individual instances?* **Likely no**, notwithstanding the aforementioned copyright restrictions.

### 4.7 Maintenance and Evolution

*Who is supporting/hosting/maintaining BookCorpus?* BookCorpus is not formally maintained or hosted, although a new version called BookCorpusOpen [16] was collected by Shawn Presser and included in the Pile [19]. As BookCorpus is no longer officially maintained, we answer the below questions in part by considering efforts to replicate and extend BookCorpus.

*Is there an erratum for BookCorpus?* **No**. To our knowledge, Zhu and Kiros et al. [49] have not published any list of corrections or errors in BookCorpus.

*Will BookCorpus be updated?* An updated version of BookCorpus is available as BookCorpusOpen [16]. This updated version was published by Presser, not Zhu and Kiros et al. [49] who created the original BookCorpus.

*Will older versions of BookCorpus continue to be supported/hosted/maintained?* BookCorpus is no longer available from the authors' website, which now tells readers to "visit smashwords.com to collect your own version of BookCorpus" [49].

*If others want to extend/augment/build on/contribute to BookCorpus, is there a mechanism for them to do so?* **Yes**, GitHub users maintain a "Homemade BookCorpus" repository [25] that includes software for collecting books from Smashwords.

*How has BookCorpus been extended/augmented?* The most notable extension of BookCorpus is BookCorpusOpen [16], which was included in "The Pile" [19] as BookCorpus2, and includes free books from Smashwords as of August 2020.

## 5 Discussion

This work provides a retrospective datasheet for BookCorpus, as a means of addressing documentation debt for one widely-used machine learning dataset. The datasheet identifies several areas of immediate concern (e.g. copyright violations, duplicated books, and genre skews), as well as other potentially

concerning areas that may guide future work (e.g. problematic content, skewed religious viewpoints, and lopsided author contributions). We suggest that BookCorpus serves as a useful case study for the machine learning and data science communities with regard to documentation debt and dataset curation. Before discussing these broader implications, it is important to note limitations of our work.

## 5.1 Limitations and Societal Impact

While this work addresses all suggested questions for a datasheet [20], it suffers some notable limitations. First, while we obtained data files for the original BookCorpus directly from the original authors' website [49], it remains unconfirmed whether these files represent one specific version of BookCorpus, when that version came to exist, and whether it was the version that other researchers used to train models like BERT. For example, some of the empty files in the dataset may correspond to books that researchers removed some time after the initial data collection. On the other hand, the files aligned perfectly with many metrics reported by the authors when introducing the dataset, so it is likely that we analyzed either the true original version of BookCorpus or a lightly-modified version.

By analyzing the original BookCorpus and the files within it, our work violates copyright restrictions for hundreds of books that should not have been distributed in a free machine learning dataset (that is, based on a plain language interpretation of their copyright statements). Given the widespread use and impact of BookCorpus, we deemed retrospective documentation necessary despite the fact that it violates copyright restrictions. However, we do not consider these violations dismissed. The machine learning community must pursue measures to reconcile with Smashwords authors, whose work was used without their consent and often in violation of their requests.

This highlights another limitation of our work, which is that it surfaces many questions it does not answer, and thus does not completely "pay off" the documentation debt incurred for BookCorpus. While our analysis aptly completes the datasheet framework and reveals important areas of immediate concern, it also highlights that some areas could benefit from additional analysis and attention. We now identify some areas in both of these categories.

## 5.2 Areas of Immediate Concern

This work found evidence of at least three areas of immediate concern with respect to BookCorpus: copyright violations, duplicate books, and genre skews. In terms of copyright violations, we found that many books contained copyright claims that, based on a plain language interpretation, should have prevented their distribution in the form of a free machine learning dataset. Many books explicitly claimed that they "may not be redistributed to others for commercial or non-commercial purposes," and thus should not have been included in BookCorpus. Also, at least 323 books were included in BookCorpus for free even though the authors have since increased the price of the book. For example, the full text of *Prodigy* by Edward Mullen is in BookCorpus (as `366549.txt`), even though the author now charges $1.99 to download the book from Smashwords [31]. To address these copyright concerns, we encourage future analysis by legal scholars to help clarify these violations and explore potential reconciliatory measures with authors.

A second immediate area of concern is the duplication of books. BookCorpus is often cited as containing 11,038 books, though this work finds that only 7,185 of the books are unique. The duplicates did not necessarily impede BookCorpus' original use case [49], however, redundant text has been a key concern for language model training datasets. The Colossal Clean Crawled Corpus (C4) [39], for example, discards all but one of any three-sentence span that occurred more than once. In many applications, future research using BookCorpus should take care to address duplicate books and sentences.

A final area of immediate concern is the skewed genre representation we identified in BookCorpus, which over-represented romance books. This skew may emerge from a broader pattern in the self-publishing ebook industry, where authors consistently find that "kinky" romance novels are especially lucrative [24, 18, 42]. In other words, because the romance genre is over-represented in the set of self-published ebooks, it is also over-represented in the subset of free ebooks.

But romance novels often contain explicit content that can be problematic in many use cases for BookCorpus, particularly when context is ignored. For example, BookCorpus contains a book called "The Cop and the Girl from the Coffee Shop" (`308710.txt`) [46], which notes in the preamble that

"The material in this book is intended for ages 18+ it may contain adult subject matter including explicit sexual content, profanity, drug use and violence." On Smashwords, the book is tagged with "alpha male," and "submissive female," and thus could contribute to well-documented gender discrimination in computational language technologies [4, 47, 6]. Little harm is done when mature audiences knowingly consume adult content, but this awareness often does not apply for text generated by language models. Thus, while the genre skew is concerning in and of itself, this example of adult content also highlights a concerning area that calls for future work.

### 5.3 Areas for Future Work

Overall there are many potential directions for research in dataset documentation, though here we note three that surfaced in our work: problematic content, skews in religious viewpoints, and lopsided author contributions. The book mentioned above, "The Cop and the Girl from the Coffee Shop," represents just one example of content that would likely impede language technology in many use cases. For example, a generative language model trained on many similar books would be susceptible to generating pornographic text and reproducing harmful gender stereotypes. Again, although the original text may have been written in good faith and consumed by informed, consenting adults, using this text to train a language model could easily reproduce similar text in vastly different contexts. Further work would be helpful to determine the extent of this potentially problematic content within BookCorpus, and its associated implications. This may involve developing tools to inspect BookCorpus and related datasets, similar to the C4 search tool introduced by [13]

Further work might also help clarify skews in religious viewpoint. Metrics from BOLD [12] suggest that some language models trained on BookCorpus favor Christianity (based on sentiment analysis), and our Smashwords21 dataset does suggest that Smashwords over-represents books about Christianity. However, the original BookCorpus dataset does not include the metadata required to precisely determine religious representation in BookCorpus. Further work might explore methods for generating such metadata, including potentially distinguishing between books *about* a given religion and books written from a particular religious viewpoint.

Finally, future work might delve further into measuring lopsided author contributions. Once again the Smashwords21 dataset hints at several points of potential concern, such as "super-authors" that publish hundreds of books. This prompts normative considerations about what an ideal "book" dataset *should* look like: which writers should these datasets contain? Should work by prolific writers be sub-sampled? If so, how? As argued by [35], machine learning research can greatly benefit from engaging such questions in pursuit of more careful, systematic dataset curation and documentation. We agree that such work can help address a range of practical and ethical challenges in machine learning research.

### 5.4 Conclusion

This paper begins to pay off some of the documentation debt for machine learning datasets. We specifically address the influential BookCorpus dataset, highlighting a number of immediate concerns and potential areas for future work. Our findings suggest that BookCorpus provides a useful case study for the machine learning and data science communities, illustrating how widely-used datasets can have worrisome attributes when sparsely documented. Some may suggest that sophisticated language models, strategic fine-tuning, and/or larger datasets can drown out any effects of the worrisome attributes we highlight in this work. However, datasets themselves remain the most "upstream" factor for improving language models, embedding tools, and other language technologies, and researchers should act accordingly. While standards and norms appear to be moving in the right direction for dataset documentation, in the meantime, post hoc efforts (like the one offered in this paper) provide a key method for understanding and improving the datasets that power machine learning.

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
