# OpenReview forum: "Addressing "Documentation Debt" in Machine Learning: A Retrospective Datasheet for BookCorpus"
_NeurIPS.cc/2021/Track/Datasets_and_Benchmarks/Round1 — NeurIPS 2021 Datasets and Benchmarks Track (Round 1)_

### Official Review · Reviewer_EXoQ · 2021-07-03

**Rating:** 6
**Confidence:** 2
**Correctness:** Correct
**Clarity:** Clear

**Strengths:**

1. Very clearly written
2. The paper addresses a very practical issue, which retrospects the large language model pretraining.

**Weaknesses:**

Not sure

**Additional Feedback:**

N/A

**Documentation:**

N/A

**Relation To Prior Work:**

clearly discussed

**Summary And Contributions:**

This paper addresses the "documentation debt" problem in BookCorpus,  a popular text dataset for training large
language models. Notably, researchers have used BookCorpus to train OpenAI’s GPT-3 models and Google’s BERT models, even though little to no documentation exists about the dataset’s motivation, composition, collection process, etc. The paper also found hints of other potential deficiencies that call for future research.

---

### Official Review · Reviewer_meAv · 2021-07-03
**Addressing "Documentation Debt" in Machine Learning: A Retrospective Datasheet for BookCorpus**

**Rating:** 6
**Confidence:** 3
**Correctness:** The observations made in the manuscri…
**Clarity:** Yes the paper is clearly written and …

**Strengths:**

The submission in its current form is very well written and is easy to follow and it brings out potential issues in one of the very popular dataset i.e., BookCorpus dataset. Some of the significant observations listed in this paper include:
1) The copyright violation of many writers in the dataset and which in turn applies to all the Language models trained on this dataset without the consents of the corresponding authors.
2) With the increasing importance of fairness in ML/AI tools, it is hard to overlook the findings in which authors point out that there are contents in the dataset which might actually exacerbate gender discrimination in the learning process.
3) The skew in genre pointed out in this work is of importance. However, I should point out that given the language models are usually trained on multiple large corpora, such skews can be compensated by usage of other corpus too. Having said that these are matters for further areas of research.
4) Finally, in summary the paper brings out a very fine and crisp datasheet in the form of a data card for BookCorpus


**Weaknesses:**

1) My suggestion to the authors can be to include some statistical significance tests in the portions where they are claiming for skew in the distributions based on Genre and religious viewpoints. Without that simply claiming the existence of a skew might not be scientifically correct.

2) Are such datasheets available for other large corpus being used by Language Models? If not, then why have the authors chosen to go ahead with the BookCorpus dataset only is not clearly mentioned. If such datasheets exist, it would be really helpful to point to some such datasheets in the manuscript.


**Additional Feedback:**

Please refer to the weakness section of the current review.

**Documentation:**

This is a datasheet created for a prior published dataset BookCorpus. Yes, the current submission elaborately discusses the documentation and has also brought out some issues with the dataset and its existing documentation.

**Ethics:**

This work brings out few ethical concerns regarding the BookCorpus dataset. The discussion is written in section 4.2.

**Relation To Prior Work:**

Yes it has been added in the draft,

**Summary And Contributions:**

In this paper, the authors are focussing on preparing a cleaner third party datasheet for BookCorpus dataset. BookCorpus dataset has been widely used for training large neural language models e.g., GPT-N, BERT and some of its variant models etc.. Given the wide usage of such language models and the embeddings learnt by them in numerous downstream tasks, the authors argue the dataset itself needs to be correctly documented and its usage should be cautiously done in future. The authors found some evidence which suggests that (a) the dataset violates the copyright restrictions of many books, (b) there are some redundant books in the dataset, and (c ) there are significant skews in the genre representations. Most worryingly, however, the authors also point to some documents which contain sexual and violent contents and thus having the potential to exacerbate gender discrimination in the learnt representations.

---

### Official Review · Reviewer_ytvn · 2021-07-04
**This type of retrospective analysis of datasets is good, but contributions are limited in scope and some claims are questionable**

**Rating:** 4
**Confidence:** 3
**Correctness:** 1. The authors anchor heavily on thei…
**Clarity:** The paper is exceedingly clear and we…

**Strengths:**

There are several things to like about this work.

1. Dataset documentation is important. The dataset in question is important, and widely used. We should encourage this type of retrospective work. It doesn’t appear that similar work has been done for the BookCorpus, so the author’s contribution is filling that knowledge gap in the community.
2. The authors are comprehensive and thorough in answering the data sheet questions.
3. The concerns raised by the authors on duplication and genre skew appear to be new contributions. I am not familiar with prior work that has identified these issues in the corpus.


**Weaknesses:**

I’m not sure if contribution of this paper is significant enough.

1. None of the types of concerns the authors raise are novel. Copyright, duplication, and “skew” issues have been raised before, for other datasets.
2. There’s little that’s methodologically novel about the analysis conducted--the authors are applying an existing framework for analyzing datasets. This particular analysis does not suggest any future considerations for how other datasets should be analyzed--the authors do not uncover a deficiency in the datasheets approach. Nor do they offer a new way of conducting these analyses.
3. Noting that the BooksCorpus skews in different ways--either by book genre or religious texts--does not itself seem to be a significant contribution.
4. Finally, the copyright declarations for the BooksCorpus have themselves been noted before [1]. At best, the authors merely quantify how common they actually are. But, there are other issues with this analysis (noted below).

[1] https://www.theguardian.com/books/2016/sep/28/google-swallows-11000-novels-to-improve-ais-conversation

**Additional Feedback:**

I think the paper engages in an interesting exercise, by retroactively creating a datasheet for a widely used dataset. As far as I can tell, the authors are the first to do this for the BooksCorpus dataset. This type of work is valuable and necessary--we should always critically reexamine old datasets with new tools.

But, several aspects of this paper make me question whether it is suited for this venue.

1. There is no methodological innovation or contribution here. The findings are largely restricted to one dataset, and do not suggest learnings that can be more broadly applied. The authors do not find--through their investigation--better ways to conduct such analyses for other datasets.

2. Much of the paper rests on claims regarding the legality of using this dataset, given the copyright declarations attached to several of the books. First, these are legal issues that lie outside the scope of this venue. Second--and more importantly--the analysis conducted in this paper is not a legal one. Given that other literature [2] suggests these issues are far more complex and nuanced, I'm not sure if the arguments made in this paper are correct. At the very least, I'd like to see reference to (1) sources of legal authority (e.g. cases), (2) consultation with legal scholars, or (3) references to legal scholarship. None of those exist in the current iteration of the paper.


**Update July 19**: I'm grateful for to the authors for their response and the changes they've made to their paper. However, I still have my original concerns and am keeping the original review score.

**Documentation:**

Yes.

**Ethics:**

The authors suggest that their analysis of the BooksCorpus may itself violate the copyright restrictions ("By analyzing the original BookCorpus and the files within it, our work violates copyright restrictions for hundreds of books that should not have been distributed in a free machine learning dataset." on Page 8). Again, I think the authors mistake the actual protections afforded by copyright laws. I'd recommend the authors consult with legal scholars--or the literature around machine learning and intellectual property--before making such strong claims.

**Relation To Prior Work:**

Yes.

**Summary And Contributions:**

The authors create a retrospective data sheet for the BookCorpus dataset. They note that the popularity of this dataset--combined with the lack of documentation when it was released--produce an increased need for such documentation. They provide answers to the questions included in a dataset data sheet.  In addition, they argue three findings: (1) the BookCorpus violates copyright restrictions, (2) it contains duplicated books, and (3) exhibits a significant skew in genre representation.

---

### Author Response · Authors · 2021-07-12
**Comments and revisions in response to initial reviews**

Thanks to the reviewers for attentive and constructive feedback in the initial reviews. We used this feedback to make several key revisions to our paper that clarify its scope, motivation, and implications, improving its overall contribution to the NeurIPS Datasets and Benchmarks Track.

Specifically, we made the following improvements:
* We agree with reviewer 2 that the paper will be strengthened by more succinctly contextualizing it relative to related work. The revised draft includes a new “Background” section that elaborates on related work mentioned in the introduction, and explains how similar studies helped motivate our documentation effort for BookCorpus.
* The “Background” section also addresses reviewer 2’s question about why we specifically chose BookCorpus: related work has provided documentation efforts for several influential web-based text datasets, but not for the influential BookCorpus dataset that has been used in many of the same models. Ultimately, this choice is in service of contributing a case study in documenting a highly influential dataset. We also edited the introduction to emphasize this point.
* Reviewer 1 raises several important points about our analysis of copyright violations. Although a full legal analysis is outside the scope of this paper, we have revised the manuscript to clarify that our analysis is based on a plain language interpretation of copyright statements, and added references to legal scholarship where appropriate (e.g. in the “Is BookCorpus distributed under a copyright…?” section). The revised manuscript also re-emphasizes the importance of future analysis by legal scholars to help clarify these violations and explore potential reconciliatory measures with authors (e.g. in the “Areas of Immediate Concern” subsection of the discussion).
* In the revised draft, we expand our discussion of “skewed” genre and religious content, addressing concerns raised by reviewer 1 and reviewer 2. We specifically clarify that the analysis does not attempt to make statistical comparisons with some kind of “natural” distribution of genres. Rather, we argue that frequency data for genres (or analogous categories in other kinds of datasets) is best used as a plain descriptive statistic that must be considered in the context of a dataset’s intended use case. We welcome more discussion on this point.

We would also like to address concerns of novelty expressed by reviewer 1. The [call for papers for the NeurIPS Datasets and Benchmarks track](https://neurips.cc/Conferences/2021/CallForDatasetsBenchmarks) includes “audits of existing datasets” and “identifying significant problems with existing datasets and their use” as example contributions within the scope of the track. The [blog post introducing the track](https://neuripsconf.medium.com/announcing-the-neurips-2021-datasets-and-benchmarks-track-644e27c1e66c) also suggests that “papers could make use of dataset documentation frameworks, such as datasheets for datasets,” noting that the track seeks “examples of known exemplary as well as problematic datasets.”

We agree with reviewer 1 that our paper’s methods and framework do not focus on novel methods of analysis (indeed, we intentionally focus on descriptive statistics about the dataset). However, we argue that the main contribution of the paper aligns with the NeurIPS Datasets and Benchmark track, as expressed in the blog post and call for papers. The revised manuscript clarifies its main contribution (namely in the abstract and introduction): a formal case study in retrospective dataset documentation that pinpoints several problems with the influential BookCorpus dataset.

Again, we thank the reviewers for helpful feedback, and look forward to further discussion.

---

### Decision · Program_Chairs · 2021-07-27

**Decision:**

Accept

**Comment:**

The paper presents a retrospective Datasheet for BookCorpus, documenting pervasive copyright issues, redundancy/duplication, and genre bias.

The primary weaknesses identified by reviewers are:
- None of the concerns raised by the Datasheet are novel issues -- copyright, redundancy/duplication, and bias are all known problems generally (although they have not been studied in relation to BookCorpus previously)
- Little methodological novelty -- they just take the Datasheet framework and apply it to a new dataset
- The work does not engage deeply enough with legal scholarship on copyright and with the impacts of the genre skew the observe.

The authors have (partially) addressed the third weakness in their revised draft  by expanding on discussions of copyright and the impacts of genre skew. Regarding the first two identified weaknesses, as the authors note, this paper is well within scope of the track. I believe there is an important contribution in papers that focus on qualitative and quantitative audits of well known datasets (even if the methods are not novel) and thus this paper still represent a valuable contribution. For this reason I recommend accepting this work. The paper would, however, be greatly strengthened by a deeper discussion of the implications of the copyright issues or the ways in which the genre skews have affected uses of this dataset. I strongly encourage the authors to make these additions to their final draft.